# Thermal Decomposition and Ceramifying Process of Ceramifiable Silicone Rubber Composite with Hydrated Zinc Borate

**DOI:** 10.3390/ma12101591

**Published:** 2019-05-15

**Authors:** Jiuqiang Song, Zhixiong Huang, Yan Qin, Xinyi Li

**Affiliations:** School of Materials Science and Engineering, Wuhan University of Technology, Wuhan 430070, China; duaojianghai@whut.edu.cn (Z.H.); yysjqspeaking@126.com (Y.Q.); 232046@whut.edu.cn (X.L.)

**Keywords:** silicone rubber, hydrated zinc borate, thermal decomposition, ceramization

## Abstract

The ceramifiable silicone rubber composite was prepared using hydrated zinc borate and kaolin as ceramifiable fillers. Effects of the hydrated zinc borate content and the combustion temperature on the properties of the ceramifiable silicone rubber composite were investigated. Thermal decomposition and ceramifying processes of the composite in a muffle furnace under air were also studied. The results showed that the density and the hardness of the composites increased as the content of the hydrated zinc borate increased from 0 to 30 phr. The tensile strength and elongation at break decreased. In addition, hydrated zinc borate decreased the decomposition temperature of the composite, whereas the residue weight under air atmosphere was increased. In the process of decomposition and oxidation of the ceramifiable silicone rubber composite in air, B_2_O_3_ was generated by the decomposition of zinc borate and participated in the formation of the residue network structure, which decreased the temperature of the ceramifying transition. The new phases, zinc aluminate (ZnO·Al_2_O_3_) and aluminum-rich mullite (9Al_2_O_3_·2SiO_2_), appeared after high-temperature thermochemical reactions. Microscopy images revealed that different structures were formed at different temperatures. The network structure of the ceramic residue became increasingly compact, and the compressive strength increased from 0.31 to 1.82 MPa with the increase of temperature from 800 to 1400 °C, which had a better protective effect on heat transfer and mass loss. The weight loss and the linear shrinkage of the ceramic residue was 37.6% and 21.9%, respectively, with the 30 phr content of hydrated zinc borate. The bending strength was improved from 0.11 to 11.58 MPa, and the compressive strength also increased from 0.03 to 1.14 MPa.

## 1. Introduction

Heat-shielding materials play an important role in the aerospace industry. When space vehicles, rockets, and missiles are flying at high speed, the surface temperature will rise sharply because of severe aerodynamic heating. Meanwhile, heat will be rapidly transferred into the interior structure at ultrahigh temperatures, resulting in damage to inner components. Hence, it is very important for heat-shielding materials to guarantee the normal operation of hypersonic vehicles [1]. In addition, heat-shielding materials can also achieve greater security and ensure public safety under fire [2,3,4]. Because the traditional polymer materials being used as cable, during a fire, the traditional polymer materials that are being used as cables would be decomposed and release toxic gas, exposing the copper wire and, overall, have a very poor fire-retardant effect. However, ceramifiable polymer composites can overcome this shortcoming. The ceramifiable polymer composites were first proposed by Hanu and his colleagues, and commercialized application was successfully achieved in 2004 [5]. The ceramifiable polymer composites, like common polymers, have excellent performance at room temperature and can form hard ceramic shells in fires and at temperature above 600 °C. In recent years, ceramifiable silicone rubber composites have developed rapidly. Ceramifiable silicone rubber composites have relatively lower heat-release rates and minimal sensitivity to external heat flux in comparison to most organic polymer composites [3,6,7]. Thermal decomposition of silicone rubber has been thoroughly studied, and good thermal resistance properties have also been demonstrated [8,9,10]. Ceramifiable fire-resistant silicone rubber composites have been studied by incorporating inorganic fillers, dispersant agent, and other additives into silicone rubber matrix [11]. Eutectic reactions take place at high temperature between mineral fillers and combustion products of silicone rubber, and then the hard, durable ceramic layer with porous structure is formed [12].

Ceramifiable silicone rubber composites are composed of silicone rubber, inorganic fillers, fibers, fluxing agents, and other functional additives, and have the same excellent properties as ordinary silicone rubber. Compared with traditional organic polymers, there are no toxic substances produced during the combustion, and the decomposition product, namely SiO_2_, can improve the flame-retardant effect [13]. There are also many studies regarding the effects of silicate mineral fillers on silicone rubber ceramization. Zhang [14,15] studied the effects of graphene nanoplatelets, montmorillonite, and silicon carbide whiskers on the properties of ceramifiable silicone rubber composites. A considerable improvement in the mechanical properties and thermal stability of the composites was obtained after surface modification of graphene nanoplatelets. A coordination effect of montmorillonite and silicon carbide whiskers caused the linear and mass ablation rates to decrease by 22.5% and 18.2%, respectively. The mechanism of ceramifying reaction has also been preliminarily studied. Cheng et al. [16,17,18] performed numerous studies on ceramifiable mica/silicone rubber, frit–mica/silicone rubber, and other silicone rubber composites. They found that the temperature would be above 800 °C if the ceramifying reactions were accomplished by eutectic reactions between mica and combustion products of organosilicon. However, the addition of frits could reduce the temperature of ceramifying reactions. Hanu et al. [5] found the ultimate strength of the resultant ceramic residue improved after combustion in air. They confirmed that the eutectic reaction and solidification process was influenced by the particle-size and chemical composition of the fillers, which resulted in the strength improvement of the ceramic residue. In addition, some fluxing agents with low softening temperature, such as glass frits, zinc borate, and zinc or borate oxide, were also introduced into silicone rubber to lower the ceramifying reaction temperature and improve the strength of the ceramic residue. Guo et al. [19] studied the effects of glass frits on the properties, microstructure, and formation mechanism of polysiloxane elastomer-based ceramizable composites, obtaining the optimum contents of the glass frit and catalytic decomposition mechanism of silicone rubber. Anyszka et al. [20] added borate oxide (B_2_O_3_) into silicone rubber composites to improve ceramifying reaction at low temperatures, although the thermal stability of the silicone rubber decreased significantly.

Many researchers have focused their attention on the properties of ceramifiable silicone rubber composite with low softening temperature fluxing agents, while ignoring the decomposition and ceramization mechanism of composites at different temperatures. Thus, in this paper, phase transition and reaction mechanism in the process of decomposition and ceramization were discussed. The effect of hydrated zinc borate, acting as fluxing agent, on the ceramifying properties of silicone rubber composite was also investigated.

## 2. Experimental

### 2.1. Materials

Commercial methyl vinyl silicone rubber (SR) was produced by Chengdu Zhonghao Chenguang Technology Co. Ltd. (Chengdu, China). The average molecular weight was 600,000–700,000. The vinyl content of the silicone rubber was 0.13%–0.18% per mole. Fumed silica (SiO_2_) with a Brunauer-Emmett-Teller (BET) surface area of 300 m^2^/g was used to improve the mechanical performance of the composites. Hydrated zinc borate (2ZnO·3B_2_O_3_·3.5H_2_O, ZB) with an average particle size of 5 μm was used as fluxing agent, lowering the ceramifying reaction temperature. The hydrated zinc borate released crystal water during 320–450 °C, and the content of crystal water was about 14% according to thermogravimetric analysis. Kaolin (Al_2_O_3_·2SiO_2_) with an average particle size of 11 μm was used as ceramifiable filler. Fumed silica, hydrated zinc borate, and kaolin were purchased from Shanghai Jingchun Bio-Chem Technology Co. Ltd. (Shanghai, China). 2,5-Dimethyl-2,5-di(*tert*-butylperoxy)hexane (DBPH) was also from Shanghai Jingchun Bio-Chem Technology Co. Ltd., China, and used as the curing agent.

### 2.2. Preparation of the Ceramifiable Silicone Rubber Composites

The ingredients were mixed on a two-roll mill with a gear ratio of 1:1.2 at room temperature. The silicone rubber was first softened, and then fumed silica, kaolin, and hydrated zinc borate were added until a homogeneous batch was obtained. Finally, the curing agent DBPH was added and processed until a visually good dispersion was achieved. The total mixing time was about 30 min. The samples were molded to platens by press vulcanizer at 170 °C with pressure of 10 MPa for 15 min. Then, the platens were put into an oven at 200 °C to air-dry for 2 h for additional vulcanization. The formulations of silicon rubber composites are given in Table 1.

### 2.3. Sample Combustion

Heat-shielding materials or cable prepared with ceramifiable silicon rubber composites are usually used in oxidation environments. Therefore, the combustion of samples (100 mm × 10 mm × 2 mm) was performed in a muffle furnace under air. In order to obtain the combustion residue at different temperatures, samples were heated from room temperature to 400, 500, 600, 700, 800, 900, 1000, 1100, and 1200 °C at a heating rate of 10 °C/min, and kept at each temperature for 30 min, and then cooled to room temperature in the muffle furnace.

### 2.4. Characterization

#### 2.4.1. Density and Shore A hardness

The density of the ceramifiable silicone rubber composites with different contents of hydrated zinc borate was measured according to Chinese Standard GB/T 533-2008 at room temperature. Firstly, the density bottle was weighed both before and after placing the sample inside. Then, the density bottle was filled with deionized water and weighed. Before weighing, the bubbles in the bottle and the water on the outer surface of the bottle needed to be removed. After that, the water and sample in the bottle were poured out. Finally, the bottle was filled with deionized water without sample and weighed again. Density was calculated using Equation (1). The reported results were the average value of five specimens.
(1)$$\rho =\rho_{w}\;\frac{m_{2}-m_{1}}{m_{4}-m_{3}+m_{2}-m_{1}},$$
where ρ, ρ_w_ are the density of the sample and the deionized water (g/cm^3^), respectively. m_1_ is the mass of the density bottle. m_2_ is the total mass of the sample and the bottle. m_3_ is the total mass of the sample, the deionized water, and the bottle. m_4_ is the total mass of the deionized water and the bottle.

The shore A hardness of the composites with different contents of hydrated zinc borate was conducted by LX-A shore hardness tester (Shanghai Jingmi Instruments Co., Ltd., Shanghai, China) at room temperature. The specimens were 6 mm thick, and the results were also the average value of five specimens.

#### 2.4.2. Tensile Strength and Elongation at Break

The tensile strength and elongation at break tests of the ceramifiable silicone rubber composites with different content of hydrated zinc borate were performed using a universal testing machine (Instron-4465, Instron Engineering Corporation, Norwood, MA, USA) according to Chinese standard GB/T 528-2009. The loading speed was 500 mm/min. The dimensions of the dumbbell samples were 115 mm (length) × 6 mm (width of narrow area) × 2 mm (thickness).

#### 2.4.3. Thermogravimetric Analysis

Thermal gravimetric analysis (TGA, STA449C/3/G, NETZSCH, Selb, Germany) was conducted to investigate the thermal stability of the samples with different contents of hydrated zinc borate under air. Then, a series of samples were heated at a rate of 10 °C/min. The relative mass loss and decomposition temperatures of the samples were recorded from room temperature to 1300 °C.

#### 2.4.4. Fourier Transform Infrared Spectroscopy

Fourier transform infrared spectroscopy (FTIR) was obtained in the range of 400–4000 cm^−1^ at a resolution of 1 cm^−1^ on a Nexus FTIR spectrophotometer (Thermo Nicolet, Waltham, MA, USA) using the KBr pellet technique for samples after being heat-treated from room temperature to 1200 °C.

#### 2.4.5. X-ray Diffraction Analysis

The crystal phases of the ceramic residue were identified by an X-ray diffraction (XRD) with a D8 ADVANCE diffractometer (Bruker, Billerica, MA, USA) with Cu Kα (λ = 0.1542 nm) radiation at a generator voltage of 40 kV and a generator current of 400 mA. The scan was conducted from a 2θ angle of 5 to 80° with a step interval of 4°.

#### 2.4.6. Morphology

The morphology of the ceramic residue after combustion at different temperatures was characterized by field emission scanning electron microscopy (FESEM, Zeiss Ultra Plus, Carl Zeiss NTS GmbH, Oberkochen, Germany).

#### 2.4.7. Weight Loss and Linear Shrinkage

The ceramic residue was obtained by heating samples up to 1000 °C for 30 min in a muffle furnace under air. The dimensions of the samples were 100 mm × 10 mm × 2 mm. Weight loss of the ceramic residue was evaluated according to Equation (2). Linear shrinkage of the ceramic residue was calculated according to Equation (3).
(2)$$W=\frac{M_{1}-M_{2}}{M_{1}}\times 100\%,$$
(3)$$L=\frac{L_{1}-L_{2}}{L_{1}}\times 100\%,$$
where W, L are the weight loss and linear shrinkage (%), respectively. M_1_ and M_2_ are the mass of the samples before and after heat treatment, respectively. L_1_ and L_2_ are the length of the samples before and after heat treatment, respectively.

#### 2.4.8. Bending and Compressive Strength

Bending and compressive strength of the ceramic residue were assessed using a universal testing machine (Instron-4465, Instron Engineering Corporation, Norwood, MA, USA). The loading speed was 2 mm/min. The compression length was 25% of the original thickness of the sample. The bending strength and compressive strength were calculated as follows:(4)$$\sigma_{f}=\frac{3p\cdot l}{2b\cdot h^{2}},$$
(5)$$\sigma =\frac{4F}{{\pi D}^{2}},$$
where σ_f_ is the bending strength (MPa). p is the load (N). l, b, and h are the span length (mm), width (mm), and thickness (mm) of the ceramic residue, respectively. σ, F, and D are the compressive strength (MPa), load (N), and specimen diameter (mm), respectively.

## 3. Results and Discussion

### 3.1. Density and Hardness of Ceramifiable Silicone Rubber Composites

Density and shore A hardness of ceramifiable silicone rubber composites with different contents of hydrated zinc borate are shown in Figure 1. The density of the composites increased from 1.305 to 1.442 g/cm^3^ with the increased content of hydrate zinc borate from 0 to 30 phr. Hydrated zinc borate has a higher density (2.67 g/cm^3^) compared with silicone rubber (1.1–1.2 g/cm^3^). The density of the composites increased due to the addition of high-density components. Meanwhile, the shore A hardness value of the SR5 composite with 30 phr hydrate zinc borate increased by 8 compared to the SR0 composite without hydrate zinc borate, which meant hydrate zinc borate reduced the flexibility of silicone rubber molecular chains.

### 3.2. Mechanical Properties of Ceramifiable Silicone Rubber Composites

The effect of the hydrated zinc borate content on tensile strength and elongation of the ceramifiable silicone rubber composites at break are shown in Figure 2. With the increased content of hydrate zinc borate, both the tensile strength and the elongation of the composites, to a certain extent, decreased at break. The tensile strength decreased from 6.94 to 5.32 MPa, and the elongation at break decreased from 531.79% to 314.34%. Consequently, the content of the hydrated zinc borate had a negative effect on the tensile strength and elongation of the composites at break. Adding hydrate zinc borate to the silicone rubber harmed the continuity of the matrix and led to the formation of weak interfacial interactions, resulting in the increment of defects. On the other hand, in the process of the composites being stretched, particles hindered the elongation of silicone rubber molecular chains, decreasing the elongation at break, which also illustrated the increase of the hardness of the composites.

### 3.3. Thermal Gravimetric Analysis (TGA)

Figure 3 shows the TG and DTG curves of six different silicone rubber composite samples and hydrated zinc borate in air. T_5_, T_max1_, T_max2_, and residue weight at 1300 °C are listed in Table 2. It can be seen that silicone rubber composites containing hydrated zinc borate underwent two stages of thermal decomposition process in air. The first stage at 332–425 °C was attributed to the decomposition of hydrated zinc borate, and the second stage at 425–625 °C was due to silicone rubber, as seen in Figure 3b. T_5_ and T_max2_ of SR1, SR2, SR3, SR4, and SR5 were lower than those of SR0. With the increased content of hydrated zinc borate, T_5_ decreased from 430 to 407 °C and T_max2_ shifted from 506.9 to 491.1 °C. Moreover, the residue weight at 1300 °C increased from 60.39% to 64.87%, as seen in Figure 3a and Table 2. The low decomposition temperature indicated the reduction of the thermal stability of the composites under air. This was mainly attributed to hydrated zinc borate, which accelerated the thermal decomposition of silicone rubber composites. The catalytic process of silicone rubber composite, by metal ions from decomposition, has already been reported in the literature [3,21]. On the other hand, the interaction between fillers and molecule chains of silicone rubber could cause the decrease of thermal stability of the silicone rubber composites [3].

Decomposition reactions of hydrated zinc borate occurred due to heat absorption during 320–450 °C, liberating water, boric acid and boron oxide, as seen in Figure 3b. T_max1_ was the maximum decomposition temperature of hydrated zinc borate. In this process, crystal water from the hydrated zinc borate was liberated and gasified, which caused the main mass loss of the silicone rubber composites. From Figure 3a, the value of the mass loss is about 14% for hydrated zinc borate, which is consistent with the theoretical water content of hydrated zinc borate (14.5%). Boron oxide, one of the decomposition products, started to soften at 325 °C and turned to liquid at 500 °C [22], which could form a perfect protective layer to cut off the pervasion of external heat and decrease mass loss, improving residue weight. With the continuous increase of temperature, the decomposition reaction would continue. The main decomposition of silicone rubber composites occurred at 425–625 °C. The macromolecular chains of silicone rubber were ruptured to form cyclic oligomers, including cyclic trimers and tetramers [23]. Cyclic oligomers were decomposed and ulteriorly oxidized at high temperature in air, producing CO_2_, H_2_O(g), and SiO_2_, which resulted in the second main mass loss of composites.

### 3.4. FTIR Analysis

Figure 4 shows the FTIR spectra of ceramic residue obtained by the combustion of silicone rubber composite (SR5) at different temperatures. A broad peak between 3300 and 3500 cm^−1^ was assigned to the vibration of hydroxyl (–OH). Hydroxyl was mainly from crystal water in the silicone rubber composite before heat treatment, and after combustion, the porous char layer tended to absorb moisture from the release of crystal water and air [24]. The FTIR peaks were assigned as follows: 2964 cm^−1^ for –CH_3_ asymmetric stretching, 1415 cm^−1^ for –CH_3_ deformation, and 1264 cm^−1^ for –CH_3_ wagging due to the vibration of methyl groups (–CH_3_) and which disappeared after combustion at 500 °C and higher temperatures, which meant that oxidation or other reactions of methyl and methylene on the side chains occurred. Characteristic absorption peaks of the Si–C bond at 815 and 800 cm^−1^ became weak after decompositions above 500 °C, which illustrated the destruction of the network structure of silicone rubber, resulting in fracture of the Si–C bond. The peaks observed at 1099 and 467 cm^−1^ were associated with the stretching and deformation vibration of Si–O bond, respectively. Hence, it inferred that the decomposition and oxidation of silicone rubber produced SiO_2_, H_2_O, and CO_2_ above 400 °C, according to the disappearance of methyl groups (–CH_3_), the fracture of Si–C bond, and enhancement of the Si–O bond. The FTIR peaks marked at 1409 and 917 cm^−1^ were characteristic absorption peaks of the B–O bond, which was from the decomposition reaction of zinc borate. The peak at 673 cm^−1^ was ascribed to the vibration of Al–O–Si groups and became strong above 800 °C, which indicated that some eutectic reactions between Al_2_O_3_ and SiO_2_ happened to form aluminosilicate [19].

### 3.5. XRD Analysis

The XRD patterns of ceramic residue of the sample SR5 are illustrated in Figure 5. A large hump appeared at around 2θ = 17–25° for residue calcinated at 500 °C, indicating the presence of amorphous SiO_2_. Amorphous SiO_2_ was generated from the decomposition of silicone rubber and might be the original additional SiO_2_. The diffraction peaks at 2θ = 16.4, 26.2, 30.9, 33.1, 35.2, 40.9, and 60.8° were assigned to mullite (3Al_2_O_3_·2SiO_2_). The diffraction peaks of zinc borate were weak and affected by mullite (3Al_2_O_3_·2SiO_2_) and amorphous SiO_2_. When the combustion temperature was 600 °C, the main phase in the mixture was still mullite from the XRD patterns, but the diffraction peak of zinc borate (3ZnO·B_2_O_3_) appeared at around 2θ = 28°, which illustrated that hydrated zinc borate had dehydrated to form zinc borate (2ZnO·3B_2_O_3_) before 600 °C. Thermal decomposition of zinc borate (2ZnO·3B_2_O_3_) produced boron oxide (B_2_O_3_) and zinc borate (3ZnO·B_2_O_3_). With the increase of temperature, the crystal structure of zinc borate was changed from a monoclinic to cubic lattice system. From the XRD pattern, cubic zinc borate (4ZnO·3B_2_O_3_) became the main phase at the temperature of 700 and 800 °C. When the combustion temperature was 900 °C, the peaks of zinc borate were weakened while the new phrase appeared, namely zinc aluminate (ZnO·Al_2_O_3_), which suggested that an interaction between zinc borate and aluminosilicate had occurred. During the temperature rise from 1000 to 1300 °C, zinc aluminate became the main phase, and some amorphous phase was also present. The diffraction peaks of aluminum-rich mullite (9Al_2_O_3_·2SiO_2_) appeared, and the amount of amorphous phase increased. According to the XRD pattern analysis, many thermal chemical reactions occurred during the process of silicone rubber composite combustion, which was consistent with the results of the FTIR spectra.

### 3.6. Microstructure of the Residue

Figure 6 gives the microstructure images of interior structure of combustion residue with 30 phr content of hydrated zinc borate at different calcined temperatures. The microscale morphology shows that filler particles were uniformly distributed in the matrix and had good compatibility with silicone rubber from the Figure 6a. After calcination at 400 °C, microcracks and -pores appeared on the surface of the composites, due to release of crystalline water of the hydrated zinc borate and scission of Si–CH_3_ of the silicone rubber, as seen in Figure 6b. When the combustion temperature reached 800 °C, silicone rubber was decomposed and crystalline water was thoroughly released, resulting in the appearance of lots of micropores, as seen in Figure 6c, and some spheres and rods appeared in the residue. These spheres and rods were formed by the melting of low-melting-point substances in the system. For instance, B_2_O_3_, from the decomposition of hydrated zinc borate, melted at approximately 500 °C and flowed at higher temperatures. Then, after being cooled to room temperature, spheres and rods were formed. With the increase of combustion temperature, zinc borate melted, which resulted in the formation of many liquid phase substances. The size of the spheres became bigger and bigger, and some of them came into contact with each other, as shown in Figure 6d. Then, after thermal treatment at 1200 °C, a large number of liquid phase substances were found to exist in ceramic residue, as seen in Figure 6e. Among these liquid phase substances, a bridged structure appeared. The original spherical structure had become an interconnected rod-like structure. As the temperature continued to rise, as seen in Figure 6f, a new crystalline phase with hexagonal columnar structure appeared in the ceramic residue, which indicated that thermal chemical reactions occurred between 1200 and 1400 °C. From the XRD pattern analysis, the new crystalline phase was considered as aluminum-rich mullite (9Al_2_O_3_·2SiO_2_).

### 3.7. Weight Loss and Linear Shrinkage of Ceramic Residue

The effect of the hydrated zinc borate content on the weight loss and linear shrinkage of the ceramic residue is shown in Figure 7. With the increased content of hydrated zinc borate, from 0 to 30 phr, the weight loss of the ceramic residue was reduced from 43.2% to 37.6% because the proportion of silicone rubber in the composite declined. Silicone rubber was decomposed and oxidized due to high temperatures in air, resulting in the release of a large number of gases, such as CO_2_ and H_2_O(g). Furthermore, with the increased loading of hydrated zinc borate, the linear shrinkage of ceramic residue increased from 15.1% to 21.9%. More liquid phase was generated by the melted zinc borate during the sintering process. The liquid products flowed at high temperature and filled the microhole. Then, during cooling, the liquid products shrunk under the action of surface tension, resulting in more compact structure as shown in Figure 6e.

### 3.8. Bending and Compressive Strength of the Residue

Figure 8 illustrates an obvious change in the bending and compressive strength of ceramic residue obtained at 1000 °C in air. With the increased content of hydrated zinc borate, from 0 to 30 phr, the bending strength was improved from 0.11 to 11.58 MPa, and the compressive strength increased from 0.03 to 1.14 MPa. The more liquid phase produced by melted hydrated zinc borate at high temperatures was beneficial for forming the more compact structure of the ceramic residue, resulting in the higher mechanical properties of the residue. In Figure 9, the compressive strength of ceramic residue (SR5) at different temperatures in air showed similar behavior. At lower temperatures, below the softening point of hydrated zinc borate, there was little liquid phase. According to the results of microstructure test, a highly porous structure existed in the ceramic residue. The structure of the ceramic residue became compact with increasing temperature, and the compressive strength increased from 0.31 to 1.82 MPa.

### 3.9. Ceramization Mechanism

According to above results of data analysis, the decomposition regularity and ceramization mechanism of the ceramifiable silicone rubber composites with hydrated zinc borate are revealed. First, crystallized water of hydrated zinc borate was released and vaporized (Equation (6)), which absorbed a great quantity of the heat and diluted oxygen on the surface of the materials, retarding thermal oxidation reactions. Then, the main chain (–Si–O–Si–) of methyl vinyl silicone rubber fractured to form cyclic depolymerization (Scheme 1) [23]. Although the energy of the Si–O bond (106.1 kcal/mol) is much higher than that of Si–C bond (77.9 kcal/mol) and C–C bond (82.6 kcal/mol) [25], the linear molecular chain of silicone rubber has good flexibility, which impels the 3d orbital electron of the Si to be involved in the formation of the ring-like oligomers (transition state, being oxidized to SiO_2_ later). Meanwhile, the side chain methyls and methylenes were oxidized to CO_2_ and H_2_O, due to the presence of oxygen (Scheme 2) [26].

With the increase of temperature, kaolin (Al_2_O_3_·2SiO_2_) was decomposed to generate mullite (3Al_2_O_3_·2SiO_2_, PDF number: 6-258) and SiO_2_ (Equation (7)), and zinc borate (2ZnO·3B_2_O_3_) was decomposed to generate 4ZnO·3B_2_O_3_ (cubic) and B_2_O_3_ (Equation (8)). B_2_O_3_, with low softening point, melted little by little, which caused kaolin and silica to bond together and form ceramic and a self-supporting structure with a certain mechanical strength. Eutectic reactions occurred among these oxides, producing new liquid and solid phases. At temperatures up to 900 °C, new mineral phases of ZnO·Al_2_O_3_ and ZnO·B_2_O_3_ were detected in the ceramic residue, indicating the presence of thermochemical interactions between zinc borate (4ZnO·3B_2_O_3_) and mullite (3Al_2_O_3_·2SiO_2_) (Equation (9)). Finally, mullite mineral (3Al_2_O_3_·2SiO_2_) continued to decompose and form aluminum-rich mullite (Equation (10)). With the increasing of combustion temperature and time, the new liquid phase would diffuse into the matrix, leading to bond formation among the components similar to akin to a “liquid bridge”, the result of which was a ceramic protective layer with compact structure, stable shape, and adequate mechanical strength. The protective layer effectively separated external oxygen and fire from the internal material, preventing further heat dispersion and mass loss.
2ZnO·3B_2_O_3_·3.5H_2_O → 2ZnO·3B_2_O_3_ + 3.5H_2_O(g)(6)
3(Al_2_O_3_·2SiO_2_) → 3Al_2_O_3_·2SiO_2_ + 5SiO_2_(7)
2(2ZnO·3B_2_O_3_) → 4ZnO·3B_2_O_3_ + 3B_2_O_3_(8)
4ZnO·3B_2_O_3_ + 3Al_2_O_3_·2SiO_2_ → 3(ZnO·Al_2_O_3_) + ZnO·B_2_O_3_ + 2SiO_2_ + 2B_2_O_3_(9)
3(3Al_2_O_3_·2SiO_2_) → 9Al_2_O_3_·2SiO_2_ + 4SiO_2_(10)

## 4. Conclusions

The effect of the hydrated zinc borate content and combustion temperature on the properties of ceramifiable silicone rubber composites was investigated. With the increased content of hydrated zinc borate from 0 to 30 phr, the density and shore A hardness increased by 10.5% and 11.7%, respectively. Nevertheless, the tensile strength and elongation at break decreased by 23.3% and 40.9%, respectively. In addition, the thermal decomposition and ceramifying process of the composites in a muffle furnace under air were also studied. Adding hydrated zinc borate as fluxing agent to the ceramifiable silicone rubber composites could significantly accelerate the process of thermal decomposition of the composites, decreasing the thermal stability of the composites. The T_5_ and T_max2_ of the ceramifiable silicone rubber composites had a certain degree of decline, while the residue weight had a significant improvement. In the process of combustion, fracture and oxidation of the molecular main chains (Si–O bond) and side chains (Si–C bond) of silicone rubber occurred, producing SiO_2_, H_2_O, and CO_2_. B_2_O_3_ was generated by the thermal decomposition of zinc borate at high temperatures and participated in the formation of the residue network structure, which decreased the temperature of the ceramifying transition. New phases, zinc aluminate (ZnO·Al_2_O_3_) and aluminum-rich mullite (9Al_2_O_3_·2SiO_2_), appeared after high-temperature thermochemical reactions. After thermal treatment above 1000 °C, a large number of liquid phase substances were found to exist in ceramic residue, and formed a bridged structure. With the increasing of temperature from 800 to 1400 °C, the network structure of the ceramic residue (SR5) became increasingly compact, and the compressive strength increased from 0.31 to 1.82 MPa, which had a better protective effect on heat transfer and mass loss. The weight loss and linear shrinkage of the ceramic residue was respectively 37.6% and 21.9%, with the 30 phr content of hydrated zinc borate. With an increased content of hydrated zinc borate, from 0 to 30 phr, the bending strength of the ceramic residue was improved from 0.11 to 11.58 MPa, and the compressive strength increased from 0.03 to 1.14 MPa.

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
