# Peer review of "Thermal Decomposition and Ceramifying Process of Ceramifiable Silicone Rubber Composite with Hydrated Zinc Borate"

_materials, 2019, doi:10.3390/ma12101591_

Round 1
Reviewer 1 Report
The manuscript authored by Song et al. treated a very interesting topic but after the reading I have some perplexities that I need to be clarified. My main concern regards the nature of degradation/thermal treatment carried out by the authors. In the abstract they wrote about pyrolysis and cited among the used techniques TG (Thermogravimetric analysis) and DSC (Differential scanning calorimetry), thus one can think that the TG was carried out under inert environment. By the contrary in the experimental the authors wrote that TG was carried out in air. Why? Why do you perform pyrolysis in a furnace? How is possible to control the inert nature of the atmosphere in a furnace? Why do you not perform pyrolysis in the thermos balance, as usual? Furthermore in the subsection sample pyrolysis the authors, among the experimental parameters did not cite the atmosphere. The doubt that there may be a misunderstanding arises spontaneously to the reader. Another concern regards the discussion about the heat release (started in the experimental and conclude in the results and discussion section). It is not possible measure the heat change of a material with TGA, probably the authors misunderstood with DSC or DTA. Again misunderstanding in the reader, because in the abstract they wrote about DSC, in the experimental DSC is not discussed/presented and in the discussion they present an anomalous plot in Figure 3b. The red one is a very strange derivative curve. Are you sure that is a DTG curve and not a DTA one? In my experience I never seen a DTG curve with two different verses. The manuscript is then affected by large number of typos and mistakes that I highlighted in the attached .pdf together with some suggestions to improve the state of the art in the introduction. Following the over reported considerations I think that the manuscript needs a strong revision before publication on Materials.

Author Response
Thank you for your comments and suggestions. The manuscript has been seriously revised according to your suggestions. The response to your questions is in the attached .docx.

Reviewer 2 Report
This manuscript describes the effects of hydrated zinc borate and the heat treatment temperature on the thermal degradation and ceramifying of silicone rubber composite. To be published in materials, the following items must be corrected or answered.
1. If the authors present the properties of the materials used in this research, it will be a great help to evaluate the results shown later. If it can be supplemented, please correct it.
2. On page 4, the meaning of F must be described as force or load (not pressure).
3. On page 7, the fourth line, "The diffraction peaks ... ... " The statements in the sentence do not seem to be general. To assert such a claim, other results that can be backed up should be presented.
4. 4. At the bottom of page 7, the last word in the fifth line should be modified to "phase" (not phrase).
5. Figures 5, 6, and 7 all lack of discussions. In most cases, they just describe the results. Authors must infer the valid reason for such results and discuss the validity of inference in conjunction with other results.
6. The results of Figures 5, 6, and 7 are unclear as to which samples are used. There is no mention about ZB contents. Although the heat treatment temperature is used as a parameter, the resultant components on the result is difficult to be considered without knowing the added components.
Author Response
Thank you for your comments and suggestions. The manuscript has been seriously revised according to your suggestions. Here is the response to your questions.
Point 1: If the authors present the properties of the materials used in this research, it will be a great help to evaluate the results shown later. If it can be supplemented, please correct it.
Response 1: The properties of the materials used in this research have a great influence on the experimental results. Properties of the raw materials, such as particle size, BET surface area and TGA parameters, has been supplemented in the manuscript.
Point 2: On page 4, the meaning of F must be described as force or load (not pressure).
Response 2: It has been revised according to your suggestion.
Point 3: On page 7, the fourth line, "The diffraction peaks ... ... " The statements in the sentence do not seem to be general. To assert such a claim, other results that can be backed up should be presented.
Response 3: According to the XRD analysis, the crystalline phases of zinc borate and boron oxide did exist, but the peak intensity was weak. A large number of silica and kaolin in the system became the main crystalline phases.
Point 4: At the bottom of page 7, the last word in the fifth line should be modified to "phase" (not phrase)
Response 4: It has been revised according to your suggestion.
Point 5: Figures 5, 6, and 7 all lack of discussions. In most cases, they just describe the results. Authors must infer the valid reason for such results and discuss the validity of inference in conjunction with other results.
Response 5: According to your suggestions, discussions about the results of these analyses have been made up in the manuscript. For example, Effect of the content of the hydrated zinc borate on the weight loss and linear shrinkage of the ceramic residue was discussed. Silicone rubber was decomposed and oxidized due to high temperature in air, then released a large number of gases, such as CO2, H2O (g), which resulted in loss of weight. On the other hand, the liquid products flowed at high temperature and filled the micro hole. Then the liquid products shrinked under the action of surface tension during cooling, resulting in more compact structure and linear shrinkage. But in order to make the manuscript look more concise, there is no duplicate discussion on the same reasons leading to different results according to the requirements of the journal.
Point 6: The results of Figures 5, 6, and 7 are unclear as to which samples are used. There is no mention about ZB contents. Although the heat treatment temperature is used as a parameter, the resultant components on the result is difficult to be considered without knowing the added components.
Response 6: Figures 5, 6 and 7 show the results of sample SR5. And it has been also supplemented in the manuscript.
Reviewer 3 Report
The method for measuring density should be described.
The parameters for mechanical testing should be described.
Was the TGA performed in air, nitrogen or inert gas?
What was the size of the specimens used for mass/dimension changes?
In lines 180-185, there is a description of what changes in chemistry happen in the TGA and although this is probably what happens, there is no evidence in this section to prove this.
Line 251: "in the composite was declined." should be "in the composite was reduced" or
"in the composite declined."
Results of bending strength are presented, but this is not described in the method section. Either include the bending method details or remove the bending results.
The authors should describe what is new in this paper compared to the paper at ICCM21: http://iccm-central.org/Proceedings/ICCM21proceedings/papers/3502.pdf
Author Response
Thank you for your comments and suggestions. The manuscript has been seriously revised according to your suggestions. the response to your questions is in the attached. docx.

Round 2
Reviewer 1 Report
I am sorry but the answers given by the authors do not solve my concerns. They continue to write about pyrolysis in air and heat release during TGA, that in my knowledge is impossible.
Author Response
Thank you for your comments and suggestions that are very important for us to improve our manuscript. The manuscript has been seriously revised according to your suggestions. We have carefully corrected the language errors in the manuscript and asked the relevant professionals to check them. In addition, we have improved the presentation of research background, test methods, results, discussions and conclusions in the manuscript.
The ceramifiable silicone rubber composites are often used in the fields of external thermal protection of hypersonic vehicles, internal thermal insulation of rocket engines and fire protection of building. Thus, we believe that it is of great practical significance to study the pyrolysis and ceramization of the composites under air. In this experiment, thermogravimetric analysis (TGA) was performed in air. Characteristic decomposition temperatures and mass changes of the composites were obtained. The heat release of endothermic decomposition mentioned in this manuscript is based on relevant literature and is used to describe the heat change in the pyrolysis. In order not to cause misunderstanding, this quotation was deleted. We hope that your questions would be answered.
Reviewer 2 Report
I have seen the answers to what I mentioned about this paper. Everything has been corrected or an appropriate response has been obtained and it appears to be available in the journal.
Author Response
Thank you for your comments and suggestions. The manuscript has been seriously revised according to your suggestions. We have improved the presentation of research background, methods, results, discussions and conclusions in the manuscript.
Reviewer 3 Report
No further comments.
Author Response
Thank you for your comments and suggestions. The manuscript has been seriously revised according to your suggestions. We have carefully corrected the language errors in the manuscript and asked the relevant professionals to check them.
Round 3
Reviewer 1 Report
You continue to associate this process to the presence of air, this is unacceptable. Pyrolysis is the thermal decomposition of materials at elevated temperatures in an inert atmosphere.
"Compendium of Chemical Terminology". International Union of Pure and Applied Chemistry. 2009. p. 1824.
Author Response
I'm very sorry for my error that the word "pyrolysis" was misunderstood. And thank you very much for your help and comments about my manuscript.
In our research, we want to study the changes of phase, microstructure, mechanical properties and mechanism of the ceramifiable silicone rubber composites when they are heated and combusted in air. Pyrolysis is the thermal decomposition of materials at elevated temperatures in an inert atmosphere,as you mentioned. So we realized that what we did was not the pyrolysis of the composites, but combustion, and revised the statement in the manuscript. We hope to have your understanding and further suggestions for our research .
Thank you very much again.